# An Intertemporal Analysis of Post-FCTC Era Household Tobacco Consumption in Pakistan

**DOI:** 10.3390/ijerph16142532

**Published:** 2019-07-16

**Authors:** Biplab Kumar Datta, Muhammad Jami Husain, Nigar Nargis

**Affiliations:** 1Global Noncommunicable Disease Branch, Division of Global Health Protection, Center for Global Health, Centers for Disease Control and Prevention, Atlanta, GA 30333, USA; 2American Cancer Society, Washington, DC 20004, USA

**Keywords:** tobacco use, tobacco control, smoking tobacco, smokeless tobacco, expenditure burden, Pakistan

## Abstract

Since the ratification of the WHO Framework Convention on Tobacco Control (FCTC) in 2004, Pakistan has taken various measures of tobacco control. This study examines how these tobacco control measures are associated with change in household-level tobacco consumption patterns in Pakistan over the decade (2005 to 2016) after FCTC ratification. We used multiple waves of the household survey data of Pakistan from 2004–2005 to 2015–2016 for analyzing household-level tobacco use. We find that tobacco consumption remains at a significantly high level (45.5%) in Pakistan despite the recent declining trend in the post-FCTC era. During the preparatory phase of FCTC implementation between 2005 and 2008, the smoking rate was on the rise, and smokeless tobacco use was declining. Over the implementation phase of FCTC policies between 2008 and 2016, the pattern of change in tobacco use reversed—the smoking rate started to decrease while smokeless tobacco use started to rise. However, the decrease in the smoking rate was slower and the increase in smokeless tobacco use at the national level was driven by an increase among the poor and middle-income households. These trends resulted in the growing burden of tobacco expenditure among the poor and middle-income households relative to the wealthier households.

## 1. Introduction

Tobacco use is a major public health issue in Pakistan. Nearly 15 million people (age 15+) consume tobacco every day, and more than 160 thousand are killed by tobacco-caused diseases every year in Pakistan [1]. About one in every five adults (19.1%) in Pakistan consumes some form of tobacco, and one in every ten adults (10.5%) is a cigarette smoker [2]. Exposure to second hand tobacco smoke at home is 21% among children of age 13 to 15 in Pakistan [3]. Mouth and oropharynx cancer and lung cancer, the two types closely related to tobacco use, are the leading causes (23%) of male cancer deaths in Pakistan [4]. The burden of diabetes and cardiovascular diseases, for which tobacco use is a key risk factor, are also high in Pakistan [5]. The tobacco epidemic in Pakistan, thus, is a serious public health concern and requires effective interventions to contain tobacco use at the population level.

Since the ratification of the World Health Organization (WHO) Framework Convention on Tobacco Control (FCTC) in 2004, Pakistan has taken various measures of tobacco control. A Tobacco Control Cell was established in the Ministry of Health (later placed under the Ministry of National Health Services, Regulations and Coordination) in 2007 to reduce the prevalence of tobacco use in Pakistan. The Tobacco Control Cell facilitates FCTC implementation through various administrative, legislative, and coordination measures, including tobacco and tobacco product taxation, a ban on tobacco product promotion and advertisement in print and electronic media, a ban on smoking in places of public work and public service vehicles, a ban on cigarette sale to minors, and anti-tobacco information campaigns. This study examines how these tobacco control measures are associated with change in household-level tobacco consumption patterns in Pakistan over the decade (2005 to 2016) after FCTC ratification. Analyzing these trends is essential for understanding the progress in tobacco control and recognizing the areas for improvement. Our research would thus inform policies for facilitating effective tobacco control programs in Pakistan.

## 2. Materials and Methods

### 2.1. Data

We used household-level data from the household income and consumption expenditure segment of the 2004–2005, 2007–2008, and 2011–2012 versions of Pakistan Social and Living Standard Measurement Survey (PSLM) and the Household Integrated Economic Survey (HIES) of 2015–2016 to analyze household-level tobacco consumption in Pakistan. PSLM and HIES are nationally representative surveys that collect information on various household consumption. These surveys provide comprehensive information on household-level income and expenditure, and the findings are used to generate important socioeconomic indicators, such as poverty incidence [6]. Both surveys ask questions about monthly household consumption of cigarettes, biri, raw tobacco, pan/betel leaves, betel nuts, and other chewing tobacco products. We classified these tobacco products in smoking and smokeless tobacco categories for analysis. The household surveys include urban and rural households from four provinces and 27 administrative divisions in Pakistan and cover 14,708, 15,512, 15,807, and 24,238 household units in 2004–2005, 2007–2008, 2011–2012, and 2015–2016, respectively.

### 2.2. Tobacco Control Measures

During the post-FCTC era, the government and public health authorities in Pakistan undertook various tobacco control measures. The effective tax rate, including excise duty and value-added tax (as percentage of price of the most sold band), increased from 52.5% in 2008 to 60.3% in 2016. The change in cigarette tax system (excise duty) from three-tiered to two-tiered structure and elimination of the ad valorem excise tax in 2013 was a major change in tobacco-related fiscal policy. As of 2016, the specific excise tax accounted for 45.8% and the value-added tax accounted for 14.5% of the retail price of the most sold brand of cigarettes. The increase in tobacco taxes contributed to increasing cigarette prices and making cigarettes less affordable in Pakistan. Between 2008 and 2016, the price of the most sold cigarette brand increased from Rs. 18.4 to Rs. 67.0 (75% increase after adjustment for inflation), and the cost of purchasing 100 packs of this brand increased from 2.9% to 4.4% of the per capita GDP of Pakistan in corresponding years [7].

Minimum compliance for making all public places completely smoke-free was achieved by 2010. Medium-size warnings in cigarette packaging with all appropriate characteristics were introduced by 2010. National anti-tobacco campaigns with 5–6 appropriate characteristics were conducted by 2014. Pakistan also enforced a ban on advertising, promotion, and sponsorship of tobacco products on national television, radio, and print media by 2014. Figure 1 provides a snapshot of the major tobacco control measures in Pakistan from 2004 to 2016.

With regard to these tobacco control measures, we investigated the following changes from 2008 to 2016 in Pakistan: (i) Share of households consuming smoking and smokeless tobacco, (ii) household expenditure shares for smoking and smokeless tobacco at the extensive margin (average of all households) and at the intensive margin (average of tobacco user households), (iii) smoking and smokeless tobacco consumption by economic status, (iv) average smoking and smokeless expenditure share (at the extensive and the intensive margins) by economic status, and (v) burden of tobacco expenditure by economic status.

### 2.3. Household Tobacco Consumption

We estimated the share of households consuming any form of tobacco (smoking and smokeless) over years for all households and separately for urban and rural households. These are the households that have at least one adult (age 15+) member using tobacco products. We reported the share of tobacco-consuming households for any tobacco and for three mutually exclusive groups—smoking only, smokeless only, and dual-use (both smoking and smokeless tobacco). Adding the shares of smoking only and dual-use, and smokeless only and dual-use generates shares of households consuming smoking tobacco and smokeless tobacco, respectively.

We then estimated the likelihood of four mutually exclusive events of no tobacco consumption, smoking only, smokeless only, or dual-use (both smoking and smokeless tobacco use) using the following multinomial logistic model:(1)$$P\left(T_{i}=c|D,X\right)=\frac{\exp\left(\beta_{0}+d_{i}^{\prime }\gamma_{t}+x_{i}^{\prime }\beta_{j}\right)}{1+\sum_{k=1}^{3}\exp\left(\beta_{0k}+d_{i}^{\prime }\gamma_{tk}+x_{i}^{\prime }\beta_{jk}\right)}$$
where *c* represents mutually exclusive tobacco consumption categories. No tobacco consumption is the base category, and three other categories are smoking tobacco only, smokeless tobacco only, and dual-use. ***D*** is a vector of dummies for the years 2008, 2012 and 2016. The year 2005 is the base year or reference year. ***X*** is a vector of covariates including share of under 5 children, share of elderly (age 65+), share of children age 6 to 14, share of male among adults (age 15+), gender of household head, highest education of household head, household size, household income per capita (constant 2008 prices), province fixed effect, and region fixed effect (urban/rural). Equation (1) was estimated separately for the full sample and for urban and rural subsamples, and standard errors were clustered at province-region level for the full sample and at province level for the urban and rural subsamples.

The exponentiated coefficients of year dummies, *exp(γ_tk_)* are the adjusted relative risk ratios (ARRR) that refer to the relative risk of consuming tobacco of type c (e.g., smoking only, smokeless only, or both smoking and smokeless) over “no tobacco” consumption in year *t*, compared to that in the base year. For example, ARRR being 0.8 for the “smoking-tobacco only” outcome in 2008 means, relative to the base year, the likelihood of consuming “smoking-tobacco only” relative to the “no tobacco” consumption outcome is 20% less likely in 2008. We compare these ARRR for different years to assess the likelihood of tobacco consumption in respective years relative to the base year.

Estimate of *exp(γ_2016,k_)* being larger and statistically different than *exp(γ_2008,k_)* means a higher likelihood of outcome *k* in 2016 compared to that in 2008. For example, if *k* is the outcome of “smoking-tobacco only” consumption, *exp(γ_2016,k_)* is 0.8 and *exp(γ_2008,k_)* is 0.7, then households are respectively 20% and 30% less likely to consume smoking-only tobacco in 2016 and 2008, relative to the likelihood of “no tobacco” consumption. Similarly, if *exp(γ_2016,k_)* is 1.2 and *exp(γ_2008,k_)* is 1.1 then households are respectively 1.2 times and 1.1 times more likely to consume “smoking only” tobacco in 2016 and 2008. A lower and statistically different ARRR for the “smoking only” outcome in 2016 will, therefore, refer to lower likelihood of consuming smoking tobacco only in 2016 than in 2008. We employed this comparison mechanism to evaluate the odds of household tobacco consumption in each successive year of observation in the post-FCTC era.

### 2.4. Tobacco Consumption by Household’s Economic Status

Next, we examined how tobacco consumption is associated with households’ economic status and how it changes over time. For this analysis, we divided the households in deciles based on per capita consumption expenditure and plotted the average share of tobacco-consuming households in each decile over consumption per capita deciles. For reporting purpose, we referred deciles 1, 2, and 3 as poor households; deciles 8, 9, and 10 as rich households; and deciles 4, 5, 6, and 7 as middle-income households. We separately plotted the household shares for any tobacco, smoking tobacco and smokeless tobacco for the survey years 2005, 2008, 2012, and 2016. We added a quadratic fit trend line for the 2005 and 2016 survey years and studied how the pattern of quadratic fit line changed during this period. A downward sloping segment of the quadratic fit line refers to relatively higher tobacco consumption incidences of households at lower deciles, while an upward sloping segment refers to the opposite. We compared the quadratic fit line of 2005 with that of 2016 to examine how they differ and whether changes in household tobacco consumption differ across economic status over time.

### 2.5. Average Tobacco Expenditure Share

We then examined how average tobacco expenditure share in the household budget changed over time. We estimated the average tobacco expenditure share at extensive margin, that is, average for all households including tobacco non-user households; and at intensive margin, that is, average for tobacco user households only. The extensive margin reflects population level impact of tobacco use on household resources, i.e., on average, how much of the household budget share in Pakistan is allocated for tobacco consumption. The intensive margin, on the other hand, reflects the impact of tobacco use on user households only and provides information on intensity of tobacco use. We reported the results separately for any, smoking, and smokeless tobacco for all households, and for urban and rural households. For extensive margin, average tobacco expenditure share is the arithmetic mean of all households’ tobacco expenditure share. For intensive margin, average expenditure share of any tobacco is the arithmetic mean of tobacco expenditure share of households consuming either or both smoking and smokeless tobacco, and average expenditure shares of smoking tobacco and smokeless tobacco are, respectively, the arithmetic mean of expenditure share on smoking tobacco of households consuming smoking only or both types of tobacco and the arithmetic mean of expenditure share on smokeless tobacco of households consuming smokeless only or both types of tobacco.

These averages, however, are not adjusted for household-level characteristics and province and regional fixed effects. To control for these attributes and to examine the adjusted differences in average tobacco expenditure share relative to the base year 2005 at the extensive margin, we estimated the following pooled cross-section regression model:(2)$$Tobshare_{i}=\alpha_{0}+\sum_{t=1}^{3}\lambda_{t}Year_{ti}+X_{i}\alpha_{1}+\alpha_{3}Urban_{i}+Province_{p}+\varepsilon_{i}$$
where *Tobshare_i_* is the tobacco expenditure share in household *i*’s budget. *Tobshare_i_* is 0 if household *i* doesn’t consume tobacco. *Yeart_i_* is a dummy that takes the value 1 if household *i* is surveyed on year *t*, and 0 otherwise. The coefficients of *Year_ti_* – *λ_2008_*, *λ_2012_*, and *λ_2016_*, provide adjusted differences of average tobacco expenditure shares for the years 2008, 2012, and 2016, respectively, relative to the base year 2005. ***X_i_*** is the vector of household-level controls, the same as those in Equation (1). *Urban_i_* is a dummy that takes the value 1 if household *i* is located in urban region and 0 if located in rural region. *Province_p_* is province fixed effect, and *ε_i_* is idiosyncratic error term. Equation (2) is separately estimated for all households and urban and rural subsamples. Standard errors are clustered at province-region level for all households and at province level for urban and rural subsamples. The adjusted differences in tobacco expenditure share for year *t* is given by Equation (3).
(3)$$E\left[Tobshare_{i}|Year_{ti}=t,X_{i}\right]-E\left[Tobshare_{i}|Year_{ti}=t_{0},X_{i}\right]=\lambda_{t}$$

A negative and statistically significant estimate (i.e., statistically different from zero) of *λ_t_* refers to decrease in average tobacco expenditure share from *t_0_* to *t*, while a positive and statistically significant estimate refers to the opposite. Since *Tobshare_i_* contains a mass of valid zero values for tobacco non-user households, we estimate a two-part model for the intensive margin analysis. In the first part, a logistic specification of Prob (*Tobshare_i_* > 0) is estimated and in the second part, a linear specification of *Tobshare_i_* is estimated for the households with positive *Tobshare_i_*. All the covariates in both parts are the same as those in Equation (3), and the model is separately estimated for all households and urban and rural households. The adjusted differences in tobacco expenditure share at the intensive margin are given by Equation (4):
(4)$$E\left[Tobshare_{i}|Tobshare_{i}>0,\;Year_{ti}=t,X_{i}\right]-E\left[Tobshare_{i}|\;Tobshare_{i}>0,Year_{ti}=t_{0},X_{i}\right]\;=E\left[P\left(Tobshare_{i}>0\right)\right]*\;\theta_{t}+E\left[Tobshare_{i}|Tobshare_{i}>0\right]*\mu_{t}$$
where, $\mu_{t}=E\left[P\left(Tobshare_{i}>0\right)|Year_{ti}=t,X_{i}\right]-E\left[P\left(Tobshare_{i}>0\right)|Year_{ti}=t_{0},X_{i}\right]$ is the average marginal effect obtained from the first part estimation, and *θ_t_* is the marginal effect obtained from second part estimation (similar to *λ_t_* in Equation (3)). The standard errors of the adjusted differences calculated from Equation (4) were obtained using the Delta method.

We also conducted an exercise for average tobacco expenditure share across households’ economic status like the one showing association between tobacco consumption prevalence and economic status. We plotted the average household tobacco expenditure share in each decile over consumption per capita deciles for each of the four survey years. We separately conducted the analysis for extensive and intensive margins. This analysis provided us with information about how households at different economic status allocate resources for tobacco consumption and how that changes over time.

### 2.6. Tobacco Expenditure Burden by Economic Status

Finally, we estimated how households of different economic status bear the share of total household tobacco consumption in the economy. For this analysis, we first estimated the total tobacco expenditure at each percentile of total household consumption per capita using appropriate survey weights for respective survey years. We then divided the total tobacco expenditure at each percentile by aggregate tobacco expenditure of all households to get the tobacco expenditure share for each percentile. Finally, we plotted the cumulative expenditure shares at each percentile over percentiles of household consumption per capita to generate Lorenz curve type diagram. A point (*x,y*) on the graph refers to *y%* of the total tobacco expenditure share being borne by the bottom *x%* of all households in respective survey years. Thus, the graph refers to a measure of unequal distribution of tobacco consumption burden across different economic status. We compared the graphs for 2005 and 2016 for any, smoking, and smokeless tobacco to examine the changes over time.

## 3. Results

### 3.1. Household Tobacco Consumption

The percentages of households consuming tobacco over time by type of tobacco consumption are presented in Table 1. As of 2016, 45.5% of all households in Pakistan had at least one person consuming any form of tobacco. The prevalence of tobacco user households with only smokers was 21.2%, with only smokeless tobacco users was 18.5%, and with dual users was 5.8%. In rural areas, nearly half of the households reported having at least one tobacco user. The prevalence estimates by type of tobacco use are higher in rural than in urban areas.

The observations at different points in time show that overall tobacco consumption has been declining consistently since 2005. The breakdown by the type of tobacco use, however, reveals important dynamics in the composition of tobacco use in Pakistan. Between 2005 and 2008, the household-level prevalence of “smoking only” tobacco use increased (by 2.0 percentage points) while the prevalence of “smokeless only” and “dual-use” tobacco use decreased (4.4 and 3.1 percentage points, respectively). After 2008, the trends reversed—the household-level prevalence of “smoking only” tobacco use started declining while that of “smokeless only” tobacco use started rising. The prevalence of “dual-use” tobacco use continued the declining trend after 2008.

The reversal of the national level trends in smoking and smokeless tobacco use since 2008 is reflected in similar trends observed among rural households. Among urban households, in contrast, the percentages of “smoking only” and “smokeless only” types did not change between 2005 and 2008 (the prevalence estimates are not statistically different). After 2008, the percentage of “smoking only” urban households started to decline while the percentage of “smokeless only” urban households plateaued. In sum, the decrease in smoking rate in Pakistan since 2008 was shared by both urban and rural households, while the surge of smokeless tobacco use was driven by the increasing use of smokeless tobacco among rural households.

In line with the trends observed since 2008, the adjusted relative risk ratios (ARRR) for consuming any form of tobacco and for smoking including dual use decreased, while the ARRR for consuming only smokeless tobacco increased (Figure 2). The ARRR for consuming smoking only tobacco gradually decreases from 1.00 in 2008 to 0.70 in 2016, and the ARRR for consuming smokeless only tobacco increases from 0.62 in 2008 to 0.77 in 2016. The ARRR for dual use of smoked and smokeless tobacco products declines from 0.57 in 2008 to 0.48 in 2016. The patterns are also similar for urban and rural sub-samples.

### 3.2. Tobacco Consumption by Economic Status

The quadratic fit line for smoking tobacco in 2005 is concave shaped, which suggests that smoking tobacco consumption gradually increased as wealth increased at lower deciles (i.e., poorer households) and gradually decreased as wealth increased at higher deciles (i.e., richer households). This concave relationship pattern changed to a downward sloping line in 2016 demonstrating a clear negative association between smoking tobacco consumption and household wealth. This shift in the pattern of smoking by household economic status was driven by increases in smoking prevalence between 2005 and 2008 followed by decreases between 2008 and 2016 in the lower five deciles, while the upper five deciles experienced continuous decline throughout 2005–2016. Overall, between 2005 and 2016, the prevalence of smoking decreased by only 5 to 7 percentage points for the poor households (bottom 3 deciles), whereas it decreased by 8 to 14 percentage points for the middle-income households (4 middle deciles) and 10 to 15 percentage points for the rich households (top 3 deciles).

For smokeless and any tobacco consumption, the quadratic fit lines shifted downward, suggesting decrease in consumption across all income strata between 2005 and 2016. However, unlike smoking tobacco consumption, the concave patterns in 2005 remained similar in 2016. A closer look at the comparison by income strata reveals the underlying dynamics. Between 2005 and 2008, prevalence of smokeless tobacco use (not mutually exclusive) decreased across all income strata. Between 2008 and 2016, on the other hand, prevalence of smokeless tobacco use (not mutually exclusive) increased in the bottom 8 deciles by 2 to 4 percentage points while decreasing in the top 2 deciles by 1 to 2 percentage points (Figure 3). Apparently, the increase in the prevalence of smokeless tobacco consumption after 2008 was driven by the growth in smokeless tobacco use among the poor and the middle-income households. The prevalence of any form of tobacco use decreased continuously between 2005 and 2016. The decrease was higher for the rich households (top 3 deciles) by 13 to 17 percentage points than that for the poor and the middle-income households by 8 to 11 percentage points (Figure 3).

### 3.3. Average Tobacco Expenditure Share

Table 2 reports the average household tobacco expenditure share by tobacco consumption type, regions, and year. Overall, tobacco expenditure accounted for 1.4 percent of household expenditure of all households including users and non-users (extensive margin) in Pakistan as of 2016. Among tobacco user households only, tobacco expenditure share was 3.0 percent (intensive margin). Average tobacco expenditure share at the extensive margin decreased by 0.33 percentage points from 2005 to 2016. For smoking tobacco, the decrease was 0.27 percentage points, whereas the decrease was only 0.07 percentage points for smokeless tobacco. The trends for any and smoking tobacco are similar for both urban and rural areas. However, in contrast to 0.11 percentage points decrease in average expenditure share of smokeless tobacco in the rural areas, no significant changes were observed in the urban areas (increased by 0.02 percentage points only).

At the intensive margin, average expenditure share of all types of tobacco (any, smoking, smokeless) increased from 2005 to 2016. The increase in average expenditure shares of smoking tobacco was higher in rural (0.28 percentage points) than in urban areas (0.13 percentage points). In contrast, for smokeless tobacco, the increase in expenditure share was higher in urban (0.23 percentage points) than in rural (0.09 percentage points) areas.

Estimates in Table 2 do not account for household-level characteristics. The adjusted differences in average tobacco expenditure share respectively between 2008, 2012, and 2016 and the base year 2005 are presented in Table 3. After adjusting for household-level attributes and regional fixed effects, the difference in average tobacco expenditure share for any tobacco between 2016 and 2005 becomes −0.21 percentage points at the extensive margin for all households, which was −0.33 percentage points without adjustments. For smoking tobacco, the adjusted differences are negative for all, urban, and rural households. For smokeless tobacco, on the contrary, the adjusted differences are not statistically different from 0, suggesting non-decreasing average expenditure share of smokeless tobacco at the extensive margin.

At the intensive margin, the adjusted differences in average tobacco expenditure share are negative for all, urban and rural households, whereas the unadjusted differences were found positive for all and rural households. The adjusted differences in average smoking tobacco expenditure are also negative for all and urban households, but not statistically different from zero for rural households. The adjusted differences in average smokeless tobacco expenditure are not statistically different from zero for all, urban, and rural households.

### 3.4. Average Tobacco Expenditure Share by Economic Status

The average tobacco expenditure shares are further analyzed by household economic status in Figure 4. The downward sloping curves fitted to the tobacco expenditure shares by household consumption per capita deciles indicate that households in lower economic status spend a larger proportion of their household expenditure on tobacco products.

At the extensive margin, the quadratic fit lines for average expenditure shares of smoking, smokeless (not mutually exclusive) and any tobacco shifted downward from 2005 to 2016. The quadratic fit line, however, changes from concave to downward sloping line for smoking tobacco, indicating greater decrease in average expenditure share for richer households. The downward shift in the quadratic fit line for smokeless tobacco is relatively small and similar to the average expenditure share of smoking tobacco. The differences are higher at higher deciles (i.e., for richer households).

More specifically, for poor households, at the extensive margin, overall tobacco expenditure share decreased by 0.2 to 0.3 percentage points, whereas for the rich households it decreased by 0.4 to 0.5 percentage points. For smoking tobacco, average expenditure share decreased by 0.2 to 0.4 percentage points for the poor households, 0.3 to 0.5 percentage points for the middle-income households, and 0.3 to 0.4 percentage points for the rich households. For smokeless tobacco, average expenditure share decreased by around 0.1 percentage points for the rich households and did not change much for the poor and middle-income households from 2005 to 2016.

At the intensive margin, average expenditure share of smoking tobacco increased by 0.1 percentage points for the bottom 2 deciles, whereas it decreased by 0.3 percentage points for the top decile. For smokeless tobacco, average expenditure share increased by 0.1 to 0.3 percentage points across all deciles. Average expenditure share of any tobacco at the intensive margin increased by 0.1 to 0.2 percentage points for the bottom 2 deciles and decreased or remained unchanged for all other deciles. As a result, the quadratic fit lines for 2005 and 2016 for smoking tobacco almost coincided, suggesting very little change over time. On the other hand, the quadratic fit line for smokeless tobacco almost parallelly shifted upward from 2005 to 2016. For any tobacco, average expenditure share increased at lower deciles (i.e., for poor households) and gradually decreased at higher deciles (i.e., for richer households) from 2005 to 2016.

### 3.5. Tobacco Expenditure Burden by Economic Status

The concentration curve shifted upward for smoking, smokeless, and any tobacco, suggesting “increasing equality” in tobacco expenditure across household income strata, which is a misnomer for “increasing burden” of tobacco consumption on the poor (Figure 5). To put it more formally, lower-income households were bearing greater burden of tobacco expenditure share in 2016, compared to that in 2005. In 2005, the bottom 30% (poor households) bore 22.8% of the total tobacco (i.e., any tobacco) expenditure of all households, which increased to 26.7% in 2016.

In general, tobacco expenditure shares increased at the lower deciles, remained mostly unchanged for middle deciles, and decreased at higher deciles. The increase in spending share of smokeless tobacco was the largest (3.8 percentage points) for the lower middle-income group (decile 4) followed by the marginally poor group (decile 3). The decrease in smokeless tobacco expenditure share, on one hand, was the largest (2.3 to 4.7 percentage points) for the richest households (deciles 10 and 9). The increase in spending share of smoking tobacco, on the other hand, was the largest (1.9 percentage points) for the poorest households (decile 1), followed by deciles 2, 4, and 3 (0.9 to 1.1 percentage points). Thus, the burden of both types of tobacco spending gradually shifted from richer households to poorer households in Pakistan during the post-FCTC period.

## 4. Discussion

This paper provides an in-depth analysis of household-level tobacco consumption in Pakistan over a decade after the country ratified the WHO FCTC in 2004. Implementation of FCTC measures and their impact on tobacco use and cessation varied across countries. Corresponding to the FCTC articles, MPOWER, a set of six tobacco control measures was introduced by the WHO to facilitate tobacco use reduction at the country-level. The highest-level implementation of the tobacco control demand-reduction measures (i.e., P, O, W, E, and R) was found significantly associated with lowering smoking prevalence in FCTC Party countries [9]. By 2014, Pakistan only attained the highest-level implementation in P (i.e., smoke-free policies) along with Saudi Arabia, Lebanon, Honduras, and several other countries. The long-term effect size (derived from the *SimSmoke* tobacco control model) of reaching the highest level of MPOWER policies for Pakistan was estimated as −0.4%, whereas it was −5.3% for Saudi Arabia, −6.9% for Lebanon, and −4.2% for Honduras [10]. This suggests that Pakistan’s tobacco control performance in terms of the impact of FCTC measures was on the lower side compared to other similar countries.

We analyzed the trend and evolution of household tobacco use from various socioeconomic perspectives, which could offer useful insights for strengthening anti-tobacco initiatives in Pakistan. Our study demonstrated opposite trends in the consumption of smoking and smokeless tobacco in Pakistan in the post-FCTC era. At the preparatory phase of FCTC implementation between 2005 and 2008, prevalence of smoking tobacco use was on the rise in rural areas and that of smokeless tobacco use was declining in both urban and rural areas, driving the respective national level trends. Over the implementation phase of FCTC policies between 2008 and 2016, the pattern of change reversed—the prevalence of smoking tobacco use started to decrease while that of smokeless tobacco use started to rise in both urban and rural areas. The smoking rate started to decrease after 2008 in urban areas as well, while smokeless tobacco use remained steady.

The combined effect of the implementation of comprehensive tobacco control measures, including tax-induced cigarette price increases from 2008 through 2016; smoke-free law in 2009; pictorial warning label in 2010; prohibition of sales of cigarettes to minors in 2011; strengthening of restrictions on tobacco advertising, promotion and sponsorship in 2014; and national media campaigns launched in 2015 and 2016 (shown in Figure 1), seem to have contributed to the downward trend in smoking prevalence observed after 2008. In contrast, the increase in smokeless tobacco use in rural Pakistan is largely attributable to inadequate or poorly implemented smokeless tobacco control policies.

First, the tobacco control legislations use the word “smoking” and do not specifically mention “tobacco” that would cover smokeless tobacco, posing a major challenge to the enforcement of tobacco control laws to smokeless tobacco. Second, Pakistan does not apply any tax on smokeless tobacco products, a policy gap that can mislead consumers to substitute smokeless tobacco use for smoking when smoked tobacco product tax and prices are increased. Tax-induced relative price change was held accountable as one of several factors that spurred the growth in smokeless tobacco use among young males in the Unites States in the 1980s [11]. One of the key recommendations of the guidelines for implementation of Article 6 of the WHO FCTC is to tax tobacco products in a comparable way to ensure that increases in taxes and prices do not result in substitution of cheaper tobacco products [12]. Third, the public health facilities are not equipped with a support system for cessation of smokeless tobacco use. A comprehensive review of the policy gaps is available elsewhere [13,14].

The decrease in the prevalence of smoking is expected to yield large public health gains among both the urban and the rural population in Pakistan. The gain would be augmented further by the reduction in dual use, as dual users can be at increased risk for nicotine addiction and dependence than single product users [15]. The increase in smokeless tobacco use among the rural population, on the other hand, is likely to offset a substantial part of this gain. The gap (10.2 percentage points) in the prevalence of “smoking only” (28.4%) and “smokeless only” (18.2%) among rural households in 2008 disappeared by 2016 (Table 1). This finding suggests the strengthening of smokeless tobacco control measures and closing the policy gap between smoking and smokeless tobacco in Pakistan, targeting specifically the rural population.

The gain in public health from the decrease in smoking, however, seems to have been uneven across household economic status. The findings of this study indicate that the decrease has been slower among poor and middle-income households compared to the rich in the top three deciles of income distribution. It suggests that the tobacco control measures in Pakistan have been less effective in reducing smoking among the lower-income households posing a major challenge to health equity. Moreover, lower-income households experienced increase in smokeless tobacco consumption that would only deepen their losses from tobacco-caused diseases, deaths, and disabilities. Tobacco control policies that are tailored to poorer individuals are, therefore, needed to protect the economically disadvantaged population from the scourge of tobacco consumption.

Relatively slower decrease in the prevalence of smoking accompanied by increasing use of smokeless tobacco induced slower rate of decrease in tobacco expenditure share in the household budget of lower-income households. By type of tobacco product, the share of household expenditure on smoked tobacco products decreased, while the share of household expenditure on smokeless tobacco remained unchanged for poorer households. At the aggregate level, the burden of tobacco expenditure shifted from richer to the poorer-income households. The disproportionately larger burden of tobacco consumption and its health and economic consequences can be catastrophic for poor households and perpetuate the state of poverty and economic inequality. It underscores the necessity to integrate tobacco control measures with national development priorities and poverty alleviation agenda.

Raising tobacco tax and prices has proven to be an effective approach to address the growing inequality in tobacco consumption and its adverse health and economic consequences. Global evidence suggests that the poor are more responsive to tax and price increases than the rich [16,17,18]. Recent price elasticity estimates for Pakistan by income groups show that the consumption of cigarette and chewed tobacco is significantly price sensitive in the lower 60 percent and price insensitive in the top 40 percent of the income distribution [8]. This is expected because the poor spend a larger proportion of income on tobacco products. A well-designed tax policy to make tobacco products less affordable while eliminating substitution of cheaper tobacco products can induce the poor cut down tobacco consumption and expenditure at a faster rate than the rich and reverse the trend of growing tobacco expenditure burden on the poor.

Previous studies on tobacco use in Pakistan were primarily focused on assessing the socioeconomic determinants of smoking [19,20,21,22] and estimating price elasticities [8,23]. None of these studies assessed how tobacco consumption has been evolving in Pakistan over the years. To the best of our knowledge, this study is the first in literature to analyze the tobacco use trends during the post-FCTC era (i.e., after 2004) in Pakistan and to examine the evolution of the association between tobacco consumption and the burden of tobacco expenditure by economic status.

This study makes an important contribution to the existing literature by analyzing multiple tobacco product use behavior (e.g., exclusive smoking and dual use of smoked and smokeless tobacco products) in contrast to examining smoking behavior in isolation from smokeless tobacco product use. Lack of sufficient evidence on smokeless tobacco use pattern and trends has been identified as a key challenge to the full implementation of WHO FCTC measures for smokeless tobacco control and the wide gap with smoking control policies among the Parties to the Convention [12]. The findings from this study will go a long way in implementing a comprehensive tobacco control plan in Pakistan covering both smoking and smokeless tobacco use and will inform smokeless tobacco control policy implementation at the global level as well. Taxing smokeless tobacco products, imposing sales restrictions, launching smokeless tobacco focused awareness campaigns, and embedding anti-smokeless tobacco information with primary healthcare consultation for relevant population groups are some policy options that can lead to curb smokeless tobacco use in Pakistan.

One major limitation of the present study is that the prevalence of tobacco use estimated in this study from household-level data on tobacco expenditure does not represent individual-level tobacco use prevalence. For example, the percentage of households reporting positive expenditure on any form of tobacco use was 45.5 percent in 2016 (Table 1), vis-à-vis 19.1 percent adult smoking prevalence in Pakistan. These two prevalence estimates are not comparable, and the trends of these measures should be interpreted separately.

Second, it makes intertemporal comparison of tobacco consumption at the household level using multiple waves of the Pakistan Social and Living Standard Measurement Survey (PSLM) and Household Integrated Economic Survey (HIES). The differences in sampling and survey design across different waves may introduce unintended bias in the measurements and estimates in this study. However, both descriptive and econometric analyses were conducted using cross-sectional survey weights (where applicable) to maintain the national representativeness of the sample in each wave of survey and the comparability of estimates over time.

Third, household-level data on tobacco consumption and expenditure are usually reported by household heads or representative adults who may be unaware of the status of tobacco consumption and expenditures of all the residents in the household, especially the youth. It can potentially create downward reporting bias in the estimates. The estimates are, however, adjusted for the age composition of household members, specifically the share of children ages 6 to 14, which can correct for this bias.

Fourth, the estimates of household-level prevalence of tobacco use do not directly consider the effect of tobacco product prices on tobacco demand due to lack of price data. However, the inclusion of province and regional fixed effects in the regression model accounts for the geographic variation in prices.

## 5. Conclusions

Tobacco consumption remains at a significantly high level in Pakistan despite the recent declining trend in the post-FCTC era. In nearly half of rural households, at least one person uses either smoked or smokeless tobacco or both. The declining trend in overall tobacco use was driven by decreasing smoking prevalence that more than offset an increasing trend in smokeless tobacco use. A balanced approach to tobacco control policy through stronger implementation of existing measures and reduction of policy gap between smoked and smokeless tobacco control measures is necessary for accelerating the progress in curbing the tobacco epidemic. The gain in public health from the decrease in smoking has been uneven, shifting the health and economic burden of tobacco use towards the poor. A pro-poor tobacco control strategy can maximize the social benefits of tobacco control by reversing this trend while contributing to the sustainable development goals of poverty reduction.

## Figures and Tables

**Figure 1 ijerph-16-02532-f001:**
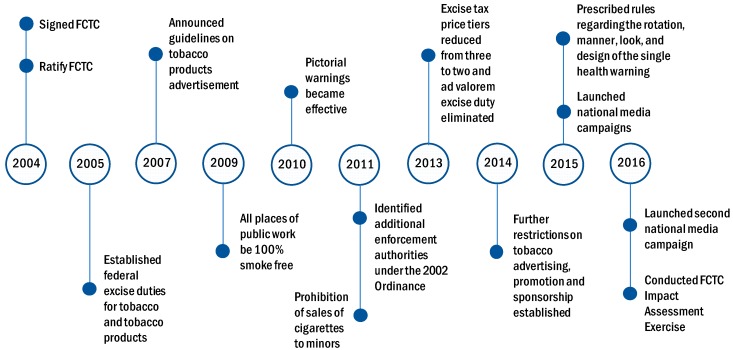
Timeline of major tobacco control measures in Pakistan. Source: Authors’ compilation from Nayab et al. (2018) [8] and Pakistan Tobacco Control Cell website.

**Figure 2 ijerph-16-02532-f002:**
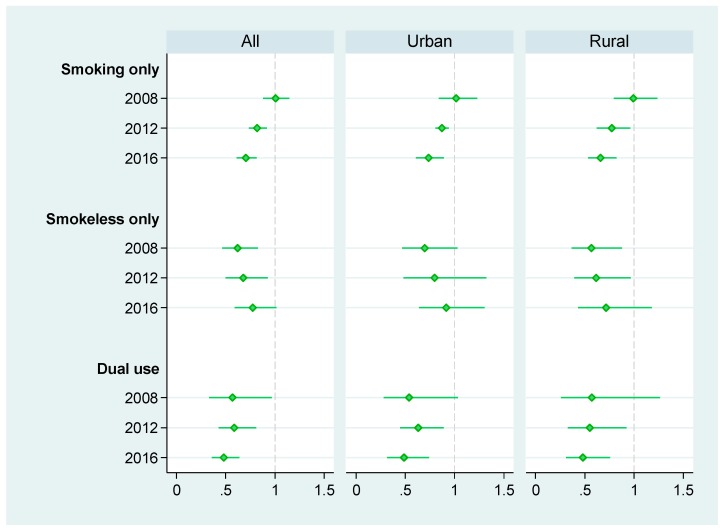
Adjusted relative risk ratios (ARRR) of mutually exclusive tobacco consumption types. Note: The solid line around the marker represents 95% confidence interval.

**Figure 3 ijerph-16-02532-f003:**
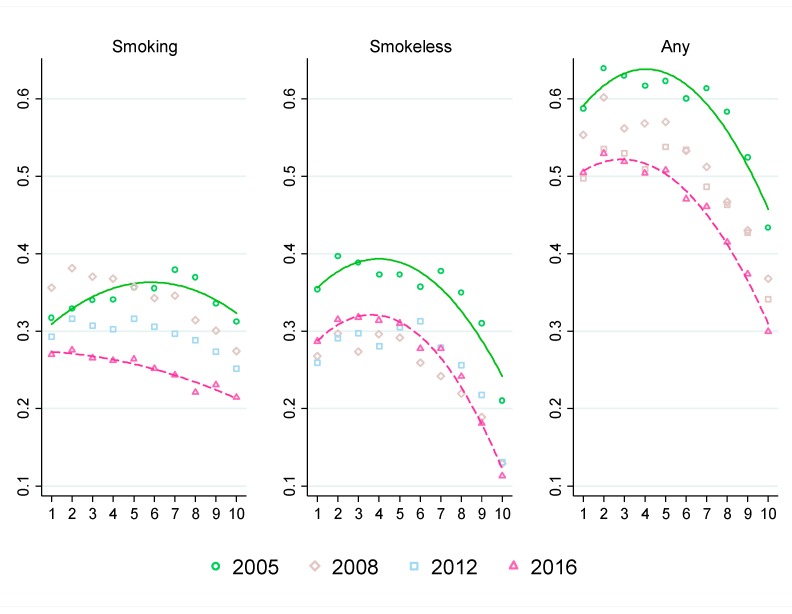
Tobacco use by consumption per capita decile and tobacco use types. Note: The vertical axes represent share (0 to 1) of households consuming tobacco. Tobacco use types are not mutually exclusive. The horizontal axes represent total household consumption per deciles. Survey weights of respective years are used to estimate household shares.

**Figure 4 ijerph-16-02532-f004:**
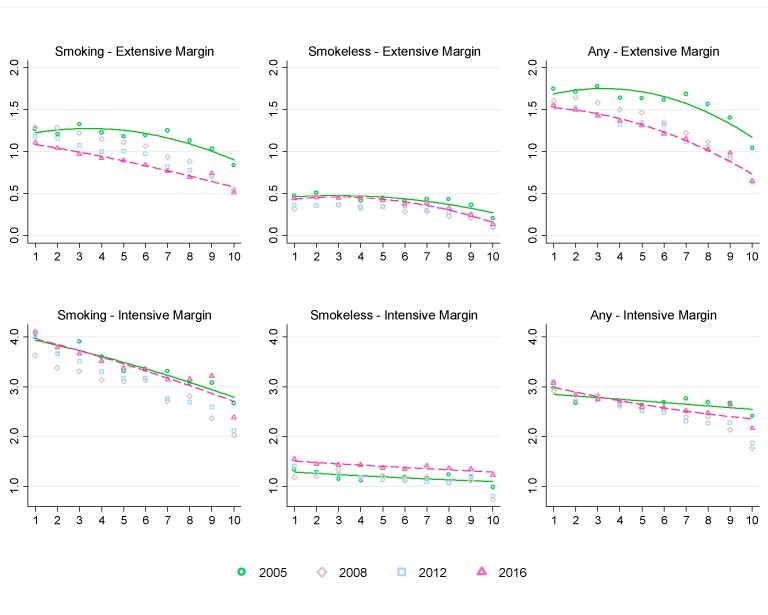
Average tobacco expenditure share by consumption per capita decile. Note: The vertical axes represent average tobacco expenditure share (%). Estimates of extensive margin involves all households including non-user households. Estimates for intensive margin only involves user households. The horizontal axes represent total household consumption per capita deciles. Survey weights of respective years are used to derive tobacco expenditure shares.

**Figure 5 ijerph-16-02532-f005:**
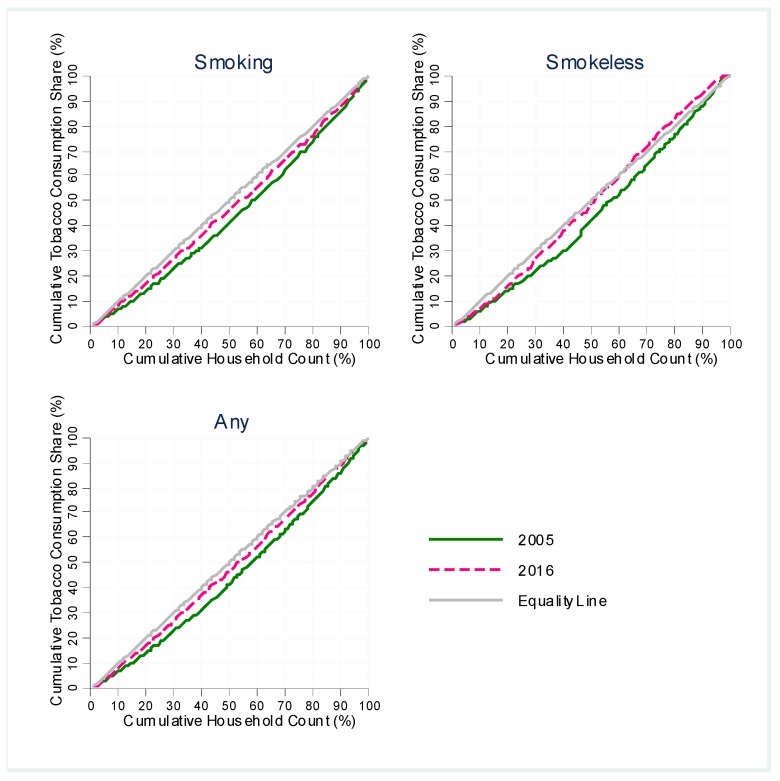
Concentration curve of tobacco expenditure share. Note: Cumulative household share presents households in ascending order of economic status. *x*% in the horizontal axis refers to poorest *x*% households.

**Table 1 ijerph-16-02532-t001:** Percentage of households consuming tobacco.

	2005	2008	2012	2016
	Percentage of Households (95% Confidence Interval)			
All Households				
Any Tobacco	57.63	52.08	49.85	45.51
	(56.05, 59.22)	(50.80, 53.37)	(48.65, 51.05)	(44.37, 46.66)
Smoking only	25.71	27.67	23.68	21.21
	(24.45, 26.96)	(26.56, 28.78)	(22.69, 24.67)	(20.36, 22.05)
Smokeless only	21.17	16.73	18.32	18.51
	(19.98, 22.37)	(15.67, 17.79)	(17.37, 19.26)	(17.61, 19.40)
Dual-use	10.75	7.68	7.85	5.80
	(9.62, 11.89)	(7.06, 8.31)	(7.19, 8.51)	(5.25, 6.35)
No. of households	14,697	15,512	15,807	24,238
Urban Households				
Any Tobacco	48.83	45.06	45.31	40.79
	(46.45, 51.20)	(43.02, 47.11)	(43.22, 47.40)	(39.53, 42.05)
Smoking only	26.13	26.30	22.92	20.19
	(24.29, 27.96)	(24.56, 28.04)	(21.41, 24.42)	(19.31, 21.07)
Smokeless only	14.43	13.84	16.25	16.37
	(12.73, 16.13)	(12.57, 15.11)	(14.58, 17.92)	(15.28, 17.45)
Dual-use	8.27	4.92	6.15	4.23
	(6.90, 9.64)	(4.18, 5.67)	(5.25, 7.05)	(3.78, 4.69)
No. of households	5798	6255	6473	16,155
Rural Households				
Any Tobacco	61.86	55.74	52.24	48.24
	(59.87, 63.84)	(54.05, 57.42)	(50.79, 53.68)	(46.58, 49.90)
Smoking only	25.51	28.38	24.08	21.79
	(23.87, 27.15)	(26.96, 29.80)	(22.79, 25.37)	(20.56, 23.03)
Smokeless only	24.41	18.24	19.41	19.74
	(22.86, 25.96)	(16.74, 19.74)	(18.27, 20.54)	(18.47, 21.02)
Dual-use	11.94	9.12	8.75	6.70
	(10.41, 13.48)	(8.24, 10.01)	(7.87, 9.63)	(5.87, 7.53)
No. of households	8899	9257	9064	8083

Note: Estimates are obtained using survey weights of respective years. The percentages of households consuming “Any Tobacco” are the sum of the percentages of households under the categories of “Smoking only”, “Smokeless only”, and “Dual-use”. Adding “Smoking only” and ”Dual-use”, and “Smokeless only” and “Dual-use” will respectively provide estimates for the percentages of households consuming “smoking tobacco” and “smokeless tobacco”.

**Table 2 ijerph-16-02532-t002:** Average tobacco expenditure share in household budget.

	2005	2008	2012	2016
**Extensive Margin**				
	Percentage of household expenditure (95% confidence interval)			
All Households				
Any	1.68	1.43	1.31	1.35
	(1.58, 1.78)	(1.37, 1.49)	(1.26, 1.37)	(1.29, 1.40)
Smoking	1.26	1.13	0.98	0.99
	(1.19, 1.34)	(1.07, 1.18)	(0.93, 1.03)	(0.95, 1.04)
Smokeless	0.42	0.30	0.33	0.35
	(0.37, 0.47)	(0.27, 0.33)	(0.31, 0.36)	(0.33, 0.38)
Urban Households				
Any	1.35	1.05	1.01	1.09
	(1.23, 1.46)	(0.98, 1.12)	(0.93, 1.09)	(1.04, 1.15)
Smoking	1.04	0.80	0.72	0.77
	(0.94, 1.14)	(0.74, 0.87)	(0.66, 0.78)	(0.73, 0.81)
Smokeless	0.30	0.24	0.29	0.32
	(0.26, 0.35)	(0.21, 0.28)	(0.24, 0.34)	(0.29, 0.35)
Rural Households				
Any	1.84	1.62	1.47	1.49
	(1.71, 1.98)	(1.54, 1.71)	(1.41, 1.54)	(1.41, 1.57)
Smoking	1.37	1.29	1.12	1.12
	(1.27, 1.47)	(1.22, 1.37)	(1.05, 1.18)	(1.05, 1.19)
Smokeless	0.48	0.33	0.36	0.37
	(0.41, 0.55)	(0.29, 0.37)	(0.33, 0.39)	(0.34, 0.40)
**Intensive Margin**				
	Percentage of household expenditure (95% confidence interval)			
All Households				
Any	2.92	2.74	2.64	2.96
	(2.78, 3.06)	(2.65, 2.83)	(2.56, 2.72)	(2.87, 3.05)
Smoking	3.46	3.19	3.11	3.67
	(3.31, 3.61)	(3.08, 3.30)	(3.01, 3.21)	(3.55, 3.80)
Smokeless	1.32	1.23	1.28	1.46
	(1.21, 1.44)	(1.14, 1.32)	(1.21, 1.34)	(1.39, 1.52)
Urban Households				
Any	2.76	2.33	2.23	2.68
	(2.57, 2.95)	(2.21, 2.45)	(2.10, 2.36)	(2.59, 2.78)
Smoking	3.03	2.58	2.48	3.16
	(2.83, 3.24)	(2.42, 2.73)	(2.33, 2.62)	(3.04, 3.28)
Smokeless	1.34	1.30	1.30	1.57
	(1.19, 1.50)	(1.19, 1.42)	(1.15, 1.44)	(1.46, 1.67)
Rural Households				
Any	2.98	2.91	2.82	3.09
	(2.80, 3.16)	(2.80, 3.03)	(2.72, 2.92)	(2.96, 3.22)
Smoking	3.65	3.45	3.40	3.93
	(3.46, 3.84)	(3.32, 3.59)	(3.28, 3.52)	(3.74, 4.11)
Smokeless	1.32	1.20	1.27	1.41
	(1.17, 1.46)	(1.09, 1.32)	(1.20, 1.34)	(1.32, 1.49)

Note: Estimates are obtained using survey weights of respective years.

**Table 3 ijerph-16-02532-t003:** Adjusted differences in average tobacco expenditure share in household budget.

	2008	2012	2016
**Extensive Margin**			
	Differences in percentage points (95% confidence interval)		
All Households			
Any	−0.243 *	−0.287 ***	−0.210 ***
	(−0.534, 0.049)	(−0.405, −0.170)	(−0.325, −0.094)
Smoking	−0.122	−0.194 ***	−0.197 ***
	(−0.331, 0.087)	(−0.304, −0.083)	(−0.294, −0.100)
Smokeless	−0.120 *	−0.094 *	−0.013
	(−0.242, 0.001)	(−0.195, 0.007)	(−0.109, 0.083)
Urban Households			
Any	−0.273	−0.273 **	−0.228 **
	(−0.817, 0.272)	(−0.439, −0.106)	(−0.405, −0.051)
Smoking	−0.199	−0.215 **	−0.257 **
	(−0.543, 0.146)	(−0.358, −0.072)	(−0.456, −0.058)
Smokeless	−0.074	−0.057	0.029
	(−0.278, 0.130)	(−0.204, 0.090)	(−0.095, 0.153)
Rural Households			
Any	−0.224	−0.308 **	−0.207
	(−0.815, 0.367)	(−0.587, −0.030)	(−0.503, 0.089)
Smoking	−0.073	−0.196 *	−0.178 *
	(−0.478, 0.333)	(−0.449, 0.058)	(−0.393, 0.036)
Smokeless	−0.151	−0.113	−0.028
	(−0.397, 0.096)	(−0.332, 0.107)	(−0.279, 0.222)
**Intensive Margin**			
	Differences in percentage points (95% confidence interval)		
All Households			
Any	−0.230 **	−0.286 ***	−0.182 ***
	(−0.407, −0.052)	(−0.415, −0.157)	(−0.294, −0.069)
Smoking	−0.082	−0.167 ***	−0.163 **
	(−0.227, 0.063)	(−0.273, −0.062)	(−0.302, −0.024)
Smokeless	−0.130 ***	−0.101 ***	−0.011
	(−0.210, −0.050)	(−0.174, −0.028)	(−0.074, 0.052)
Urban Households			
Any	−0.264 **	−0.262 ***	−0.206 ***
	(−0.476, −0.053)	(−0.394, −0.130)	(−0.312, −0.101)
Smoking	−0.16	−0.186 ***	−0.223 ***
	(−0.360, 0.039)	(−0.271, −0.101)	(−0.371, −0.074)
Smokeless	−0.095 *	−0.075 *	0.015
	(−0.194, 0.003)	(−0.164, 0.013)	(−0.061, 0.091)
Rural Households			
Any	−0.225	−0.334 ***	−0.185 *
	(−0.496, 0.046)	(−0.551, −0.118)	(−0.395, 0.025)
Smoking	−0.027	−0.175 *	−0.131
	(−0.244, 0.189)	(−0.372, 0.023)	(−0.388, 0.125)
Smokeless	−0.160 **	−0.120 **	−0.025
	(−0.287, −0.032)	(−0.234, −0.007)	(−0.153, 0.103)

Note: *** *p* < 0.01, ** *p* < 0.05, * *p* < 0.1. Each row presents coefficient estimates of year dummy variables from a single equation for extensive margin and marginal effect estimates from two-part estimation for intensive margin. Other control variables (not reported here) include share of under 5 children, share of elderly (age 65+), share of children age 6 to 14, share of male among adults (age 15+), gender of household head, highest education of household head, household size, household income per capita (constant 2008 prices), province fixed effect, and region fixed effect (urban/rural).

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
