# Peer review of "An Intertemporal Analysis of Post-FCTC Era Household Tobacco Consumption in Pakistan"

_ijerph, 2019, doi:10.3390/ijerph16142532_

Round 1

Reviewer 1 Report

I was pleased to review this interesting and well-presented paper.  My comments and suggestions are minor.

1.       The abstract would benefit from the inclusion of headline figures from the study.

2.       I did notice a number of incorrect uses of tense in the abstract and manuscript.  I would encourage the authors to review the paper and amend errors in tense use.

3.       Please spell out the first use of the abbreviation for the WHO [Page 1, Line 38].

4.       The last two paragraphs of the introductory section feel a little out of place.

a.        The sentences in Lines 5-10 on page 2 may be better suited for inclusion in the discussion and methods section respectively. 

b.       The second paragraph [Page 2, Lines 12-18] feels more suited the methods section.

5.       The reader would benefit from labelling columns containing the values and CI’s in the tables.  While I appreciate that the authors have provided this information in notations, it would benefit the flow of the paper if the reader were able to establish this at a glance.

6.       The discussion is interesting and makes pertinent recommendations.  However, it would benefit from making mention of how Pakistan compares to other similar countries in terms of the impact of FCTC measures.  

Author Response

Point 1: The abstract would benefit from the inclusion of headline figures from the study.

Response 1: Thank you for your comments and suggestions. We edited the manuscript accordingly. The household level tobacco-use prevalence rate is included in the abstract.

Point 2: I did notice a number of incorrect uses of tense in the abstract and manuscript.  I would encourage the authors to review the paper and amend errors in tense use.

Response 2: We went through the paper and edited the language as appropriate.

Point 3: Please spell out the first use of the abbreviation for the WHO [Page 1, Line 38].

Response 3: The WHO is spelled out as suggested.

Point 4: The last two paragraphs of the introductory section feel a little out of place.

a.        The sentences in Lines 5-10 on page 2 may be better suited for inclusion in the discussion and methods section respectively. 

b.       The second paragraph [Page 2, Lines 12-18] feels more suited the methods section.

Response 4: The respective texts are moved in the Methods and Discussion sections as suggested.

Point 5: The reader would benefit from labelling columns containing the values and CI’s in the tables.  While I appreciate that the authors have provided this information in notations, it would benefit the flow of the paper if the reader were able to establish this at a glance.

Response 5: The columns in the tables are labelled as suggested.

Point 6: The discussion is interesting and makes pertinent recommendations.  However, it would benefit from making mention of how Pakistan compares to other similar countries in terms of the impact of FCTC measures.  

Response 6: The following text is added in the discussion:

Implementation of FCTC measures and its impact on tobacco-use and cessation varied across countries. The highest-level implementation of the tobacco control demand-reduction measures (i.e., P, O, W, E, and R) was found significantly associated with lowering smoking prevalence in FCTC Party countries [14]. By 2014, Pakistan only attained the highest-level implementation in P (i.e., smoke-free policies) along with Saudi Arabia, Lebanon, Honduras, and several other countries. The long-term effect size (derived from the SimSmoke tobacco control model) of reaching the highest level of MPOWER policies for Pakistan was estimated -0.4%, whereas it was -5.3% for Saudi Arabia, -6.9% for Lebanon, and -4.2% for Honduras [15]. This suggest that Pakistan’s tobacco control performance in terms of the impact of FCTC measures was on the lower side compared to other similar countries.

Reviewer 2 Report

The manuscript is quite understandable in relation to the format and content that is properly explained with the various graphs and statistics. The results that allow us to know the relationship between tobacco consumption at the household level and its relationship with cultural and economic factors are also of interest. However, as the authors already mention: "... .One major limitation of the present study is that the prevalence of tobacco use estimated in this study from household level data on tobacco expenditure does not represent individual-level tobacco use prevalence". It would also be interesting to propose actions at a practical level in public health based on these results and the prevalences by ages and geographic areas that are available. Some discussion is already explained, but for example, the authors could specify a little more about the implementation of a comprehensive tobacco control plan in Pakistan that covers both the consumption of tobacco and the consumption of smokeless tobacco, especially in the population most disadvantaged.

Author Response

 Point 1: The manuscript is quite understandable in relation to the format and content that is properly explained with the various graphs and statistics. The results that allow us to know the relationship between tobacco consumption at the household level and its relationship with cultural and economic factors are also of interest. However, as the authors already mention: "... .One major limitation of the present study is that the prevalence of tobacco use estimated in this study from household level data on tobacco expenditure does not represent individual-level tobacco use prevalence". It would also be interesting to propose actions at a practical level in public health based on these results and the prevalences by ages and geographic areas that are available. Some discussion is already explained, but for example, the authors could specify a little more about the implementation of a comprehensive tobacco control plan in Pakistan that covers both the consumption of tobacco and the consumption of smokeless tobacco, especially in the population most disadvantaged.

Response 1: Thank you for your comments and suggestions. The following text is added in the discussion:

Taxing smokeless tobacco products, imposing sales restrictions, launching smokeless tobacco focused awareness campaigns, embedding anti-smokeless tobacco information with primary healthcare consultation for relevant population groups are some policy options that can lead to curb smokeless tobacco-use in Pakistan.